# Influence of Kinematic Parameters of Carbon Dioxide Turbine on Its Coefficient of Efficiency and Dimensions

**Sergey Osipov, Olga Zlyvko, Igor Miliukov, Kirill Berdyugin and Dmitriy Lvov ***

Department of Innovative Technologies of High-Tech Industries, National Research University "Moscow Power Engineering Institute", 111250 Moscow, Russia
* Correspondence: lvovdd@mpei.ru

**Abstract:** To reduce carbon dioxide emissions into the environment, the energy sector develops oxygen-fuel energy cycles. One of the most promising cycles is the Allam cycle that features the highest efficiency of electricity generation among all others. One of the main elements of an oxy-fuel energy cycle is a high-temperature carbon dioxide turbine. The turbine's working fluid and coolant consist predominantly of carbon dioxide at a supercritical pressure. Currently, there are no recommendations in the literature for the design of carbon dioxide turbines for an oxy-fuel energy system (OFES) operating according to the Allam cycle; therefore, there is a need to study the influence of parameters of the flow path of carbon dioxide turbines on its efficiency and overall performance. In this paper, we have presented the results of one-dimensional calculations of a flow path of the carbon dioxide turbine for the Allam cycle with a capacity of 300 MW, with an initial temperature and pressure of 1100 °C and 30 MPa, and an outlet pressure of 3 MPa. The study was carried out by varying the rotor speed, the reactivity level and the average diameter. Based on the results of one-dimensional calculations, we have found that the highest efficiency of the turbine flow path is achieved at a speed of 471 rad/s, a reactivity of 0.5, and an average diameter of 1.1 m for the first stage.

**Keywords:** turbine; carbon dioxide; Allam cycle; axial turbine





## 1. Introduction

In recent years, energy consumption by mankind has increased markedly, resulting in an increase in carbon dioxide emissions into the environment from combusting hydrocarbon fuels at thermal power plants [1]. Many scientists believe that it is the increase in the $CO_2$ concentration in the atmosphere that is the cause of global climate change [2,3]. Oxy-fuel energy cycles are currently being developed to produce electricity with near-zero emissions. Depending on the design type, the content of carbon dioxide in the working fluid can be from 40% to 95–97%. Depending on the working fluid flows distribution, oxy-fuel energy cycles can be open, semi-closed, or closed. The following power generation methods are widely known in this area: semi-closed oxy-fuel combustion combined cycle (SCOC-CC), MATIANT cycles, Allam cycles, and Graz cycles [4,5].

The Allam cycle has the highest efficiency among all oxy-fuel energy cycles. According to various estimates, the net efficiency of this cycle ranges from 50 to 59%. A feature of the Allam cycle is integrating heat from an air separation unit (ASU) into the circuit and its use in a multi-flow heat exchanger. The Allam cycle is a gas turbine cycle with carbon dioxide burial. The working fluid is a mixture of carbon dioxide containing water vapor with a mass concentration of about 3%. A schematic thermal diagram of the Allam cycle is shown in Figure 1.

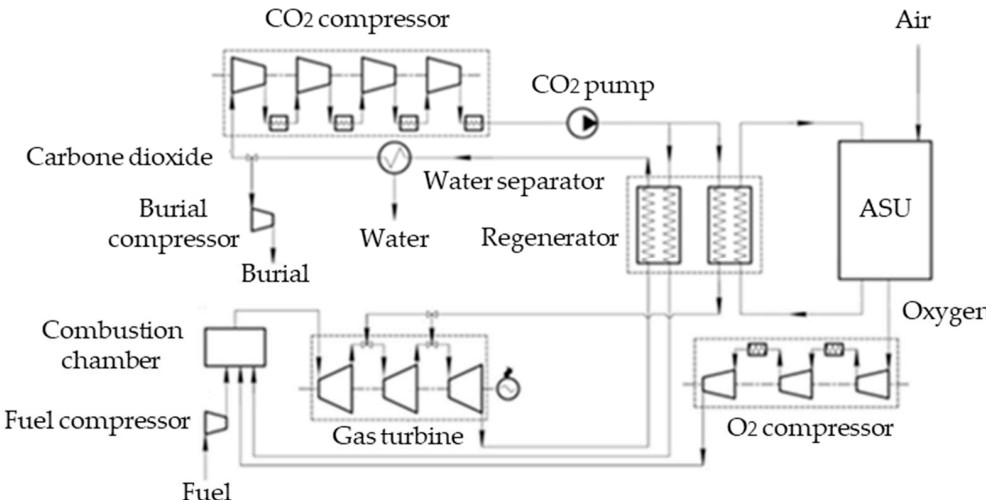

**Figure 1.** Schematic diagram of a power plant operating according to the Allam cycle.

Air is supplied to the cryogenic ASU, from which oxygen is separated and sent to the combustion chamber (CC). Methane in a stoichiometric ratio with oxygen is supplied to the CC, where it is burned, generating a mixture of carbon dioxide and water vapor. A circulating flow of carbon dioxide is also sent to the CC. The circulating carbon dioxide, mixed with the methane and oxygen combustion products, is sent to the gas turbine (GT). The resulting working fluid at a temperature of 1373.15 K and a pressure of 30 MPa drives the GT rotor. Downstream of the GT, the working fluid is supplied to the regenerator (RG), where it gives up its heat to the flow of cold $CO_2$ fed to the combustion chamber, and to the $CO_2$ flow to cool the GT blade system. Downstream of the regenerator, the cold working fluid flow is sent to the cooling separator (CS), where water vapor is separated from it. At the CS outlet, the $CO_2$ purity is up to 99%, depending on the separator used and the working fluid flow cooling degree. After the CS, carbon dioxide splits into two flows; one flow is sent for burial, and the second one remains in the cycle, being sent to the intercooled compressor suction inlet. After the compressor, carbon dioxide is sent to the pump, where pressure is finally pumped up to working parameters. The high-pressure carbon dioxide is sent to the combustion chamber where it cools the CC walls and mixes with the combustion products. The heat released by the ASU compressors is removed to a multi-flow heat exchanger (regenerator) through an intermediate air circuit. The multi-flow heat exchanger in this design is presented as a combination of two double-flow heat exchangers.

Air is supplied to the cryogenic ASU, from which oxygen is separated and sent to the combustion chamber (CC). Methane in a stoichiometric ratio with oxygen is supplied to the CC, where it is burned, generating a mixture of carbon dioxide and water vapor. A circulating flow of carbon dioxide is also sent to the CC. The circulating carbon dioxide, mixed with the methane and oxygen combustion products, is sent to the gas turbine (GT). The resulting working fluid at a temperature of 1373.15 K and a pressure of 30 MPa drives the GT rotor. Downstream of the GT, the working fluid is supplied to the regenerator (RG), where it gives up its heat to the flow of cold $CO_2$ fed to the combustion chamber, and to the $CO_2$ flow to cool the GT blade system. Downstream of the regenerator, the cold working fluid flow is sent to the cooling separator (CS), where water vapor is separated from it. At the CS outlet, the $CO_2$ purity is up to 99%, depending on the separator used and the working fluid flow cooling degree. After the CS, carbon dioxide splits into two flows; one flow is sent for burial, and the second one remains in the cycle, being sent to the intercooled compressor suction inlet. After the compressor, carbon dioxide is sent to the pump, where pressure is finally pumped up to working parameters. The high-pressure carbon dioxide is sent to the combustion chamber where it cools the CC walls and mixes with the combustion products. The heat released by the ASU compressors is removed to a multi-flow heat

exchanger (regenerator) through an intermediate air circuit. The multi-flow heat exchanger in this design is presented as a combination of two double-flow heat exchangers.

One of the most important elements of a power plant operating according the Allam cycle is a high-temperature carbon dioxide turbine. Currently, there is a prototype carbon dioxide turbine for the Allam cycle with a capacity of 50 MW that has been developed by the Japanese company Toshiba [6]. The turbine includes 7 stages; the flow path with a constant root diameter is made in a two-housing design; the inner housing is used to reduce the pressure drop and to increase the heating and cooling rate of the housing parts. The combustion chamber is installed in the same housing with the turbine.

The study [7] presents a sketch of the flow path for a 400 MW Allam cycle carbon dioxide turbine. This turbine has 9 stages and a rotor with a decreasing root diameter. The height of the working blade of the first stage is 45 mm, the last one, 136 mm; the root diameter of the first stage is 0.9 m, the root diameter of the last stage is 1.2 m. There are no more open publications presenting the results of developing an Allam cycle turbine.

The studies [8,9] present the results of developing a carbon dioxide turbine for the Allam cycle with a capacity of 335 MW, which has seven stages. The pressure and the temperature upstream of this turbine are 1083 °C and 30 MPa, respectively. The blade height of the turbine's first stage is 30 mm, the last blade length is 170 mm. The flow path of these turbines with a root diameter value is 870 mm.

Currently, there are many studies on the development of turbines for other oxy-fuel energy cycles, the working fluid of which is supercritical carbon dioxide. In [10,11], the authors described a 156 MW gas turbine flow path for the SCOC-CC oxygen-fuel cycle. They suggested two versions of the flow path: single-shaft and two-shaft designs. Both versions have a decrease in the root diameter. With a single-shaft design, the flow path includes 5 stages, and the rotor rotates at a frequency of 5200 rpm. The blade height of the first stage is 45 mm, the blade length of the last stage is about 350 mm. The first 3 stages are equipped with a cooling system since the initial $CO_2$ temperature upstream of the turbine is 1400 °C. The second version of the flow path is a two-shaft design. The first shaft has two working cascades; it rotates at a speed of 7200 rpm and is designed to drive the compressor. The root diameter of the first shaft is 1000 mm, the height of the first blade is 60 mm. The second shaft includes 3 steps; it rotates at a speed of 4500 rpm and is designed to drive an electric generator.

The study [12] presents the flow path of a carbon dioxide turbine for a power plant operating according to the GRAZ cycle. The gas turbine capacity is 618 MW. The turbine has a two-shaft design. The first rotor speed is 8500 rpm due to the fact that it is used to drive the $CO_2$ compressor. The second rotor is connected to an electric generator and rotates at a speed of 3000 rpm. The first stage height is 100 mm with a root diameter of 1066 mm. The root diameter of the last stage is 2600 mm, the blade height is 750 mm.

In [13,14], the authors developed a four-stage carbon dioxide axial turbine for the Brayton $CO_2$ recompression cycle with a capacity of 10 MW and a rotation speed of 27,000 rpm. The initial flow temperature upstream of the turbine is 700 °C, the initial pressure is 250 bar, and the pressure downstream of the turbine is 80 bar. The turbine includes 4 stages; its rotor is designed with a constant root diameter. The turbine efficiency is about 85%.

The study [15] describes the results of designing a three-stage axial carbon dioxide turbine with a capacity of 10 MW for the Brayton cycle with supercritical $CO_2$. The initial pressure upstream of the turbine is 10 MPa, the initial temperature is 500 °C, and the pressure downstream of the turbine is 9.5 MPa. The nozzle and working cascades profiles of the S-$CO_2$ turbine were calculated using the ANSYS BladeGen software. The average diameter from the first stage to the last one varies from 212 mm to 249 mm. The authors could achieve the turbine's relative internal efficiency of 91.6%.

The study [16] presents a turbine configuration consisting of two parts: a compressor turbine and a power turbine for the oxy-fuel combined cycle. The overall isentropic efficiency of the two-stage compressor turbine is 86.7%, the shaft power is 86.5 MW, and

the rotation speed is 5700 rpm. The power turbine includes four stages. The radii upstream of the power turbine are 622 mm and 740 mm, the radii downstream of the turbine are 766 mm and 1195 mm. The overall length is 980 mm, the rotation speed is 3000 rpm.

The study [17] presents the design of an axial turbine with a capacity of 450 MW for the supercritical carbon dioxide Rankine cycle. The turbine is divided into high- and low-pressure parts; both parts have a double-flow design. The rotor speed is 3600 rpm. The average rotor diameter is 660.4 mm, the first blade height of the high-pressure part is 71.12 mm, and the last blade length of the low-pressure part is 137.16 mm.

In the study [18], the consequences of some design parameters in the design of carbon dioxide turbomachines is examined. The influence of the number of stages, the speed of rotation of the rotor on the efficiency of the turbine, and its axial dimensions was studied. It should be noted that the authors analyzed these dependences for both carbon dioxide turbomachines and jet engines. An increase in the number of stages in a turbine is associated with an increase in its efficiency, but the length of the turbine also increases, which increases the capital costs of its production.

The review of the literature revealed that there are several studies devoted to developing the Allam cycle turbine. The authors mainly study turbines for oxy-fuel cycles, Brayton recompression cycles, and supercritical Rankine cycles. For the Allam cycle turbine, the authors did not conduct multifactor studies of the influence of the flow path parameters, such as the reactivity, the rotor speed, the average diameter of the first stage, the efficiency of turbine's flow path, and its dimensions. In this paper, we present the results of studies carried out for a carbon dioxide turbine power plant operating according to the Allam cycle. Having analyzed the parameters that the researchers presented in their studies, we can say that, in terms of the pressure drop, the number of stages, and the flow path shape, carbon dioxide turbines are closer to a steam turbine than to a gas one, but the temperature upstream of the turbine is closer to a gas one. Thus, when designing, it is necessary to use to a large extent the methods of designing steam turbines, but it is also necessary to take into account the design features of high-temperature gas turbines, which are primarily associated with high temperatures of working environments and the presence of a cooling system.

This paper is devoted to determining the influence of the degree of reactivity, rotational speed and average diameter on the efficiency, and the design parameters of the flow part of a 300 MW carbon dioxide turbine for an Allam cycle-based power plant. The novelty of the obtained research results consists in establishing the dependences of the influence of the average diameter of the first stage, the reactivity of the stages on the efficiency, and dimensions of the flow part of a 300 MW carbon dioxide turbine power plant based on the Allam cycle.

## 2. Materials and Methods

A one-dimensional calculation was carried out using the method for calculating steam turbines. The algorithm for this calculation is shown in Figure 2. The calculation was carried out for various rotational frequencies of 314, 471, and 628 rad/s (50, 75, 100 Hz), flow path shapes (constant root, average, and peripheral diameters), average diameters (0.7; 0.8; 1; 1.2 m), and reactivity values (0.1 to 0.7).

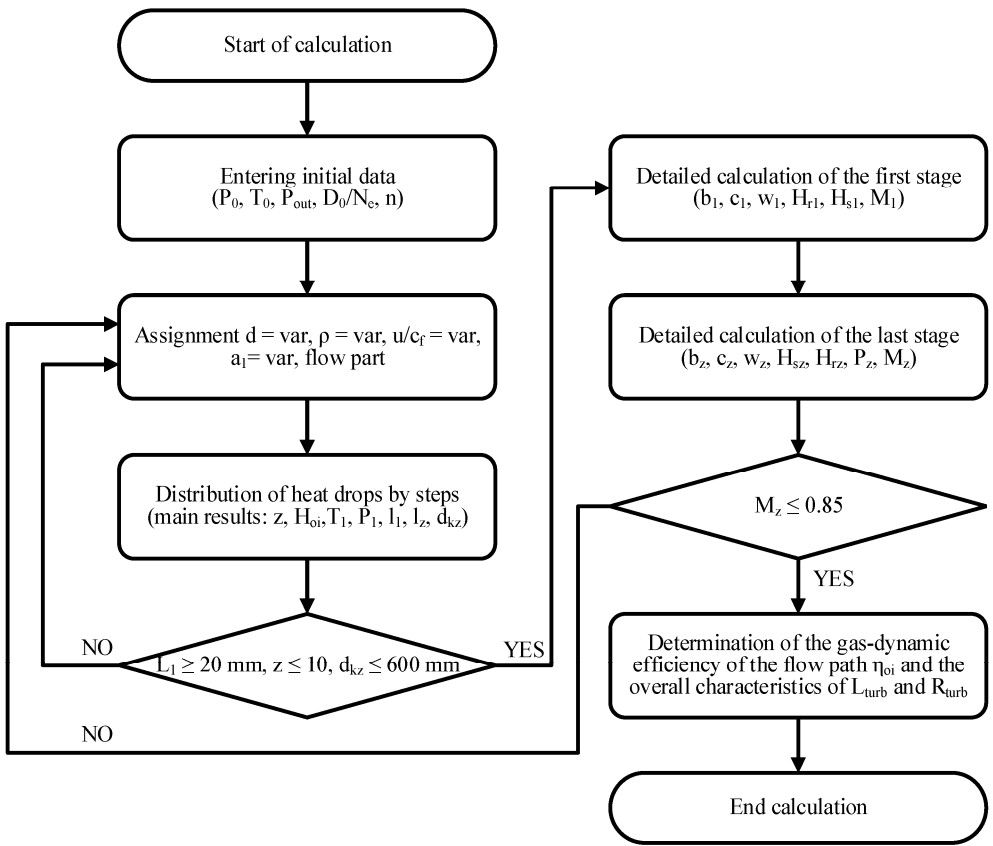

**Figure 2.** Algorithm for designing a carbon dioxide turbomachine.

The initial parameters of the thermal circuit for a power plant based on the Allam cycle are presented in Table 1.

**Table 1.** Parameters for calculating a one-dimensional gas turbine.

| Initial Pressure $p_0$, MPa | Final Pressure $p_k$, MPa | Initial Temperature $t_0$, K | Electric Power $N_e$, MW | Reactivity (for the First Stage) | Average Diameter, m (for the Second Stage) |
| --- | --- | --- | --- | --- | --- |
| 30 | 3 | 1373.15 | 300 | 0.3 | 1.1 |

In the one-dimensional calculation of a gas turbine, we made the following assumptions:

(1)　We determined the losses in the blade system of stages based on empirical dependencies for losses obtained for steam turbine profiles [19];

(2)　we did not take into account the losses due to cooling the blades of high-temperature stages;

(3)　we evaluated the bending and tensile stresses for a blade without internal cooling channels;

(4)　and the working fluid consists entirely of carbon dioxide.

During the design, we applied the following limitations:

(1)　The number of stages must not exceed 10;

(2)　the height of the first nozzle blade must be at least 20 mm;

(3)　the Mach numbers in the nozzle system of the last stage must not exceed 0.85;

(4)　and the minimum root diameter is 0.6 mm.

A one-dimensional calculation of various turbine versions was carried out subject to maintaining the optimal ratio $u/c_f$. The $u/c_f$ ratio was determined using Equation (1):

$$\frac{u}{c_f} = \frac{\varphi \cdot \cos a_1}{2 \cdot \sqrt{1 - \rho}} \tag{1}$$

where $\varphi$ is the speed coefficient in the nozzle cascade, $\rho$ is the degree, $a_1$ is the exit angle from the nozzle cascade. Figure 3 shows the turbine speed triangles.

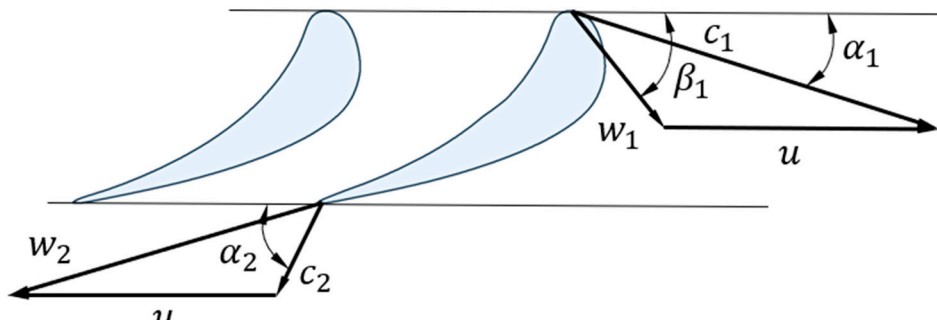

**Figure 3.** Turbine speed triangles.

Upon increasing the average diameter of the first stage, the number of stages decreases. This is due to the fact that, upon increasing the average diameter, the rotor's rotational velocity increases. To maintain parameter $u/c_f$ unchanged, we need to increase the enthalpy drop generated by each stage, which results in decreasing the number of stages.

The limitation associated with the blade height is justified by the fact that, with too small a blade height, there is a significant increase in the final energy losses. The blade height can be calculated using Equation (2) [20]:

$$l_1 = \frac{G \cdot v_{1t}}{\mu_1 \cdot c_{1t} \cdot \pi \cdot d_{av} \cdot \sin\alpha_1} \qquad (2)$$

where $G$ is the working fluid flow rate, $v_{1t}$ is the theoretical specific fluid volume, $\mu_1$ is the speed coefficient in the cascade, $c_{1t}$ is the absolute theoretical exit speed from nozzles, $d_{av}$ is the average diameter, and $\alpha_1$ is the exit angle from the nozzle cascade.

At the first stage of the calculation of a high-temperature carbon dioxide turbine, the constant parameters of pressure $P_0$ and temperature $T_0$ at the turbine inlet, outlet pressure $P_{out}$, working medium flow $D_0$ and turbine power $N_e$ were set. The variable parameters are the average turbine diameter $d$, reaction $\rho$, and rotation speed $n$.

At the second stage of the one-dimensional calculation, the main parameters of the first and last stage, and turbine are determined: the number of stages $z$, enthalpy drop per stage, pressure parameters $P_1$ and temperature $T_1$ at the outlet of the 1 nozzle stage, the height of the blades of the first $l_1$ and the last stage $l_z$, and the hub diameter $d_{hub}$. In addition, at this stage, the fulfillment of the conditions described above was checked: $l_1 \geq 20$ mm and $z \leq 10$, $d_{hub} \leq 600$ mm.

The third stage of the one-dimensional calculation is a detailed calculation of the geometric and kinematic characteristics in the first and last stages, such as: the chord of the blades $b_1$ and $b_z$, the absolute velocity $c_1$ and $c_z$, the relative velocity $w_1$ and $w_z$, the heat transfer to the nozzle and working blades of the first and last stages $H_{w1}$, $H_{n1}$, $H_{wz}$, $H_{nz}$, and the Mach numbers at the output from the first and last stage $M_1$, $M_z$. Additionally at this stage, the condition for fulfilling the condition $M_z \leq 0.85$ is checked.

At the fourth stage, the main dimensions of the turbine were determined: the radial size $R_{turb}$ and axial $L_{turb}$, and the internal efficiency of the turbine $\eta_{oi}$.

## 3. Results and Discussion

*3.1. Studying the Influence of the Average Diameter on the Flow Path Efficiency and the Dimensions of a Carbon Dioxide Turbine within the Rotation Frequency Range of 314 to 628 rad/s*

A one-dimensional calculation was carried out at a constant root diameter. This is due to the fact that a turbine with a constant average diameter always has more stages than a turbine with a constant root diameter at the same average diameters of the first stage. The fact is that in a turbine with a constant root diameter, the average diameter

increases from the first stage to the last one, which allows to generate a larger enthalpy drop in each subsequent stage, even with a small increase in the value of $(u/c_f)_{opt}$ due to increased reactivity. In a turbine with a constant average diameter, the available enthalpy drop decreases from stage to stage because the circumferential velocity is constant, whereas the value of $(u/c_f)_{opt}$ increases due to an increase in reactivity.

Figures 3 and 4 show, respectively, the results of studying the average diameter influence on the design and gas-dynamic characteristics at various rotor speeds and the dependence of the flow path efficiency on the average diameter and the rotor speed.

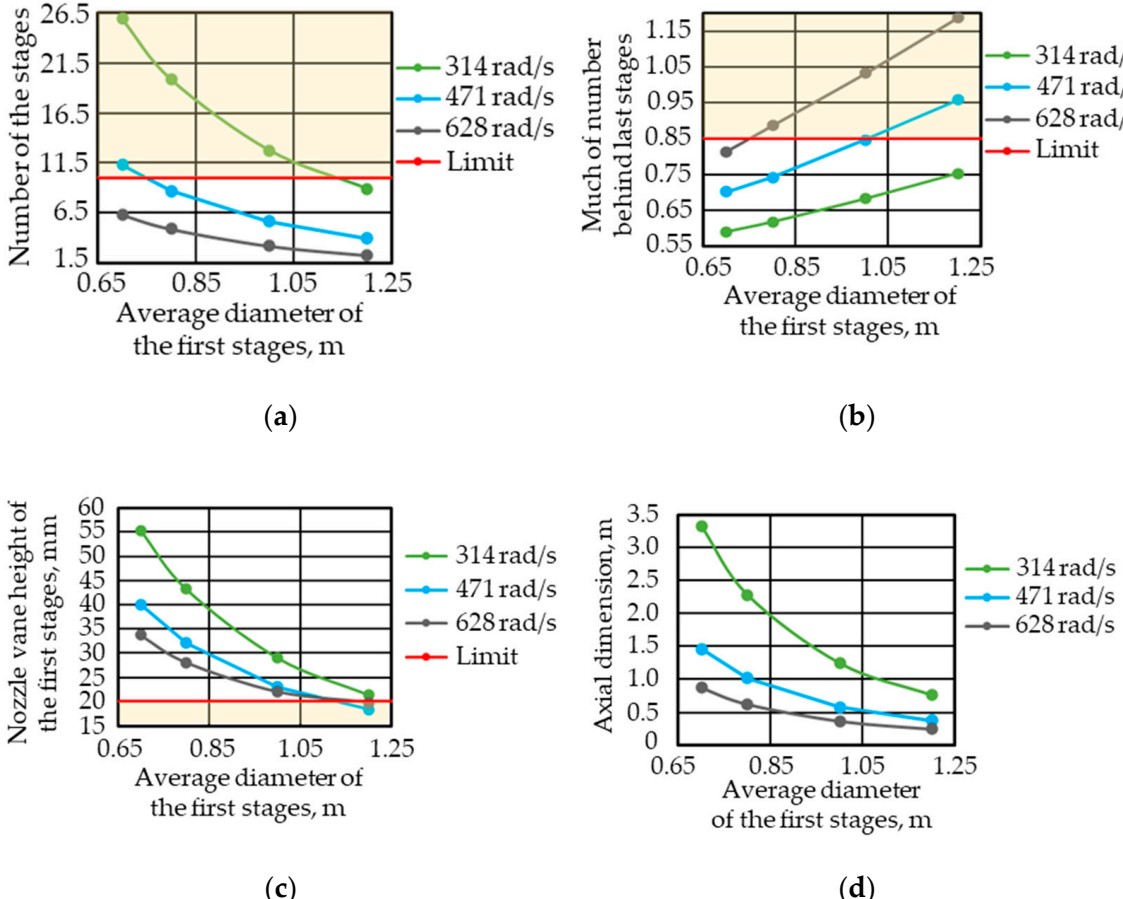

**Figure 4.** Results of studying the influence of the average diameter on the design and gas-dynamic characteristics at various rotor speeds. (**a**) Dependence of the number of stages on the average diameter. (**b**) Dependence of the Mach number downstream of the last stage on the average diameter. (**c**) Dependence of the height of the first nozzle blade on the average diameter. (**d**) Dependence of axial dimensions on the average diameter.

Our analysis of the average diameter (Figures 4–6) influence on various parameters of the flow path with various frequencies shows that at a turbine speed of 314 rad/s, it is possible to provide the number of stages less than 10, the nozzle blade height of the first stage more than 20 mm, and the Mach number in the last stage less than 0.85 with an average diameter of the first stage within a range of 1.1 to 1.2 m. In this case, the internal relative efficiency of the flow path will be 85 to 85.2%, and its axial dimension will be 0.76 to 0.95 m.

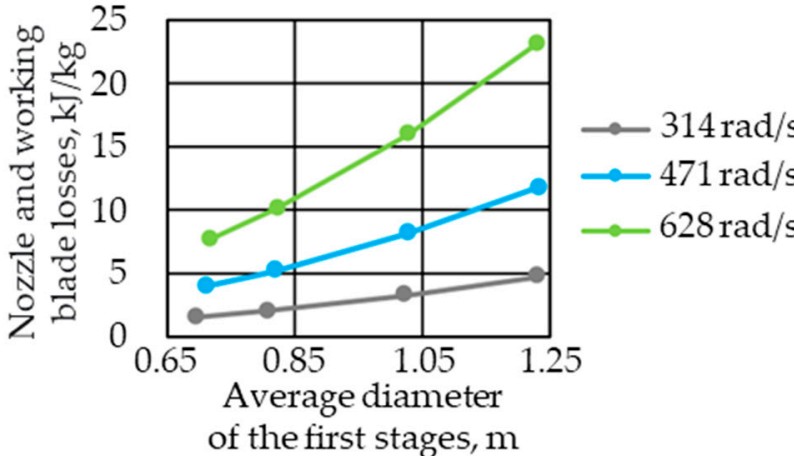

**Figure 5.** Results of studying the influence of the average diameter on the design and gas-dynamic characteristics at various rotor speeds.

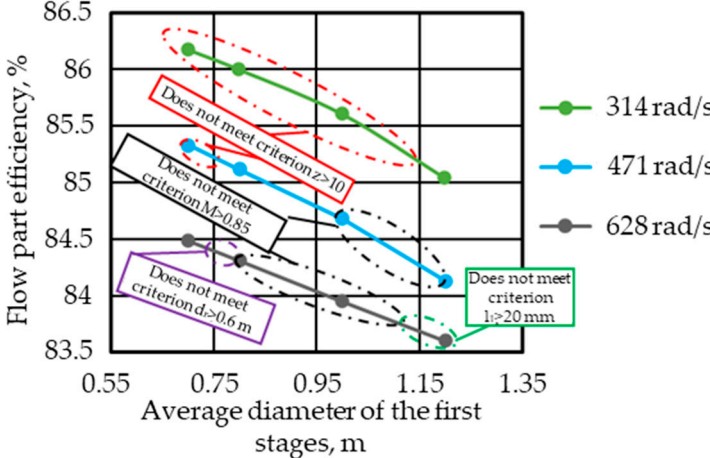

**Figure 6.** Dependence of the flow path efficiency on the average diameter and the rotor speed.

At a turbine speed of 471 rad/s, it is possible to provide the number of stages less than 10, the nozzle blade height of the first stage more than 20 mm, and the Mach number in the last stage less than 0.85 with an average diameter of the first stage within a range of 0.75 to 1.0 m. In this case, the internal relative efficiency of the flow path will be 84.7 to 85.2%, and its axial dimension will be 0.578 to 1.24 m.

At a turbine speed of 628 rad/s, it is possible to provide the number of stages less than 10, the nozzle blade height of the first stage more than 20 mm, and the Mach number in the last stage less than 0.85 with an average diameter of the first stage within a range of 0.7 to 0.75 m. In this case, the internal relative efficiency of the flow path will be 83.7 to 84.5%, and its axial dimension will be 0.75 to 0.876 m

It was also found that with an increase in the average diameter from 0.8–1.2 m, there is an increase in losses on the nozzle and working blades by an average of 3 times at different rotor speeds. This can be explained by the fact that with an increase in the average diameter of the blade chord b remains unchanged, since the tensile forces acting on the working blade and depending on the rotor speed do not change, while the length l of the blade decreases. According to the empirical dependencies presented in [19], the greater the l/b ratio, the greater the losses on the nozzle and working blades. In addition, it was found that an increase in the rotor speed also leads to an increase in losses on the working blades, which can be explained by an increase in its chord, due to an increase in the tensile forces acting on the working blades.

The results of our study show that designing a carbon dioxide turbine with a frequency of 628 rad/s is impractical since the internal relative efficiency of its flow path will be 0.7 to 1.0% less as compared to a turbine with a frequency of 314 and 471 rad/s. The turbine flow path efficiency for the 314 and 471 rad/s branches is approximately the same, being equal to 85.2%. However, at a rotation frequency of 471 rad/s, the axial dimension of the turbine flow path will be 30% smaller than that of a turbine with a rotation frequency of 314 rad/s at equal efficiency. Based on the above facts, we can conclude that designing a carbon dioxide turbine with a rotation speed of 471 rad/s will provide the maximum level of gas-dynamic efficiency with the smallest weight and dimension characteristics.

*3.2. Studying the Influence of Reactivity on the Flow Path Efficiency of a Carbon Dioxide Turbine within the Rotation Frequency Range from 314 to 628 rad/s*

Figures 7 and 8 show, respectively, the results of studying the influence of reactivity on the design and gas-dynamic characteristics at various rotor speeds and an average diameter of 1.1 m and the dependence of changing the flow path efficiency on the reactivity with an average diameter of 1.1 m.

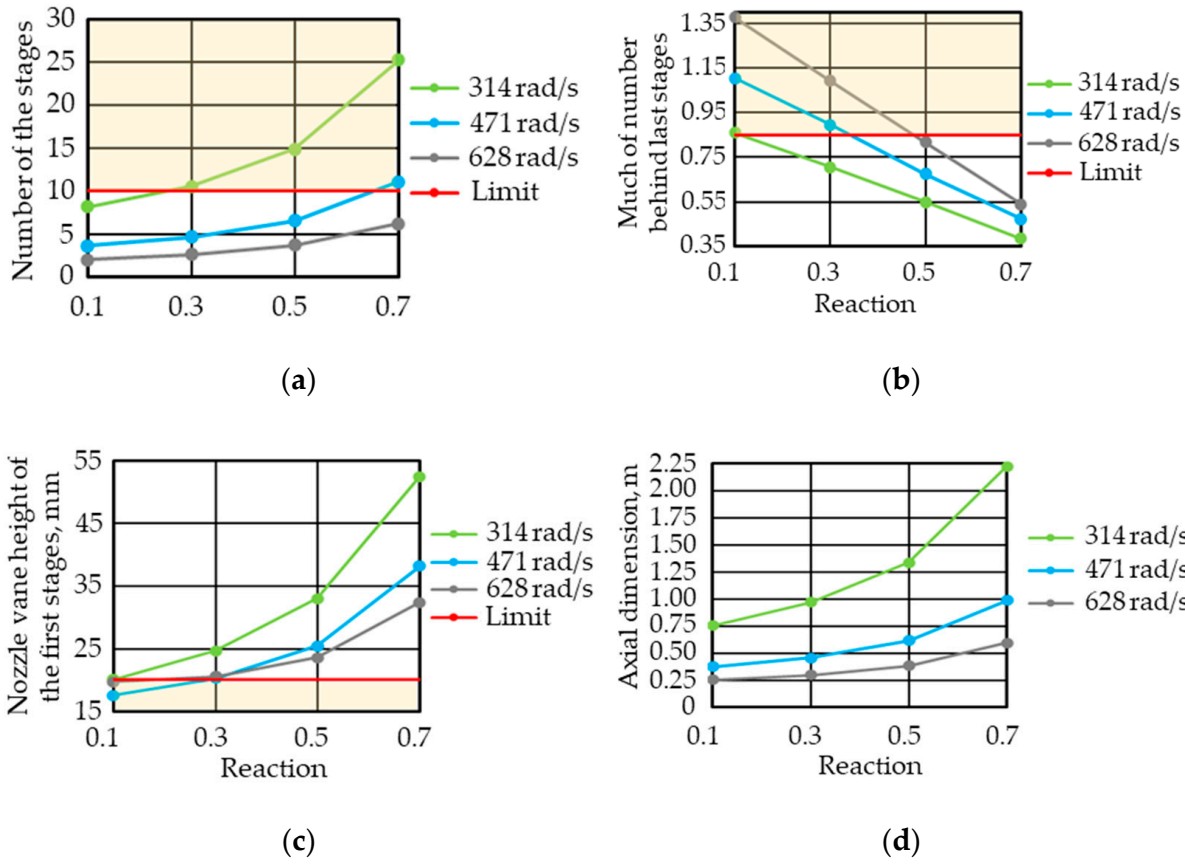

**Figure 7.** Results of studying the influence of reactivity of the design and gas-dynamic characteristics at various rotor speeds with an average diameter of 1.1 m. (**a**) Dependence of the number of stages on reactivity. (**b**) Dependence of the Mach number downstream of the last stage on reactivity. (**c**) Dependence of the first nozzle blade height on reactivity. (**d**) Dependence of axial dimensions on reactivity.

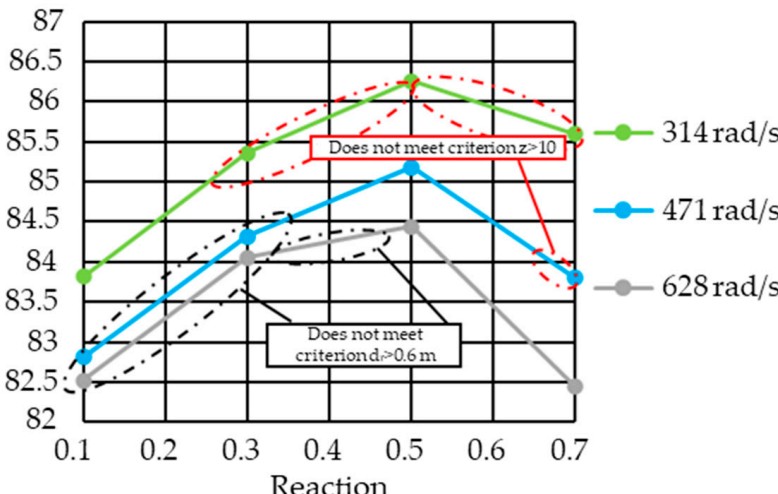

**Figure 8.** Dependence of change the flow path efficiency on reactivity with an average diameter of 1.1 m.

Our analysis of the average diameter influence on various parameters of the flow path with various frequencies shows that at a turbine speed of 314 rad/s, it is possible to provide the number of stages less than 10, the nozzle blade height of the first stage more than 20 mm, and the Mach number in the last stage less than 0.85 with a reactivity within a range of 0.1 to 0.25. In this case, the internal relative efficiency of the flow path will be 83.9 to 85.0%, and its axial dimension will be 0.75 to 0.94 m.

At a turbine speed of 471 rad/s, it is possible to provide the number of stages less than 10, the nozzle blade height of the first stage more than 20 mm, and the Mach number in the last stage less than 0.85 with a reactivity within a range of 0.35 to 0.65. In this case, the internal relative efficiency of the flow path will be 84.5 to 85.2%, and its axial dimension will be 0.524 to 0.922 m.

At a turbine speed of 628 rad/s, it is possible to provide the number of stages less than 10, the nozzle blade height of the first stage more than 20 mm, and the Mach number in the last stage less than 0.85 with a reactivity within a range of 0.475 to 0.7. In this case, the internal relative efficiency of the flow path will be 84.3 to 82.2%, and its axial dimension will be 0.426 to 0.593 m.

Table 2 presents our preliminary recommendations for designing a carbon dioxide turbine with various rotation speeds.

**Table 2.** Recommendations for designing carbon dioxide turbine for the Allam oxy-fuel energy cycle.

| Design Feature | Parameter | | |
|---|---|---|---|
| Rotation frequency, rad/s | 50 | 75 | 100 |
| Flow path shape | Constant root diameter | | |
| First blade height of the nozzle system, mm | 20 | 25 | 23 |
| Average diameter, m | 1.1 | 1.1 | 1.1 |
| Reactivity | 0.25 | 0.5 | 0.525 |
| Axial dimensions, m | 0.968 | 0.614 | 0.383 |
| Radial dimensions | 1.17 | 1.17 | 1.17 |
| Turbine's flow path efficiency, % | 85 | 85.1 | 84.4 |

## 4. Conclusions

Based on the results of our one-dimensional calculation of the flow path for an Allam cycle carbon dioxide turbine, we have obtained dependences that allow us to assess the influence of the rotation frequency, the average diameter, and the turbine reactivity on

the gas-dynamic efficiency of the flow path and the number of stages of a carbon dioxide turbine. The obtained dependences can be used for thermal calculations of the flow parts of carbon dioxide turbines for power plants based on the Allam cycle.

We have found that in order to provide the maximum efficiency equal to 85.1%, it is advisable to choose the flow path with a constant root diameter, a rotation frequency of 471 rad/s, a reactivity of 0.5, and an average diameter value of 1.1 m. With these parameters, the axial dimension of the flow path will be 0.614 m with the number of stages z = 7.

We have also developed recommendations for designing turbines with a rotation frequency of 314 and 628 rad/s. For a turbine with a frequency of 314 rad/s, it is advisable to choose its average diameter equal to 1.1 m, an average reactivity of 0.25, which will provide the flow path efficiency of 85%, the axial dimension of 0.968 m with the number of stages equal to 10. For a turbine with a frequency of 628 rad/s, it is advisable to choose its average diameter equal to 1.1 m, an average reactivity of 0.525, which will provide the flow path efficiency of 84.4%, the axial dimension of 0.383 m with the number of stages equal to 4.

The above recommendations are preliminary since they have been obtained using the exiting methods for calculating a stream turbine. Further studies will include updating the calculation method adopted to the carbon dioxide turbine operating conditions:

(1) Take into account the influence of the multicomponent composition of the working fluid and the presence of coolant flows on the thermodynamic features of the expansion process and the thermohydraulic characteristics of inter-blade channels. For this necessary:

(2) Based on the calculation and experiment studies, develop adjustments to the dependences of gas-dynamic losses of the profile cascades on the main design parameters of the flow path.

**Author Contributions:** Conceptualization, S.O. and O.Z.; methodology, K.B. and I.M.; software, S.O. and D.L.; validation, S.O. and I.M.; formal analysis, O.Z. and K.B.; investigation, I.M. and S.O.; resources, K.B. and I.M.; data curation, I.M. and S.O.; writing (original draft preparation), D.L.; writing (review and editing), D.L. and S.O.; visualization, D.L.; supervision, O.Z.; project administration, I.M. and O.Z.; funding acquisition, I.M. and O.Z. All authors have read and agreed to the published version of the manuscript.

**Funding:** This study conducted by the Moscow Power Engineering Institute was financially supported by the Ministry of Science and Higher Education of the Russian Federation (project no. FSWF-2020-0020).

**Institutional Review Board Statement:** Not applicable.

**Informed Consent Statement:** Informed consent was obtained from all subjects involved in the study.

**Data Availability Statement:** Some or all data and models that support the findings of this study are available from the corresponding author upon reasonable request.

**Conflicts of Interest:** The authors declare no conflict of interest.

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
