# Peer review of "Influence of Kinematic Parameters of Carbon Dioxide Turbine on Its Coefficient of Efficiency and Dimensions"

_inventions, doi:10.3390/inventions7040105_

Round 1

Reviewer 1 Report

Review report of inventions-2007187

Influence of Kinematic Parameters of Carbon Dioxide Turbine on Its Efficiency and Dimensions

The present study probed into the influence of kinematic parameters of carbon dioxide turbine on its efficiency and dimensions using one-dimensional method. The major concern for ensuring the Journal quality is the description of the research method which is not sufficiently clear. The theories with their relevant equations involving in the present one-dimensional analytical method must be provided and explained in a systematic manner. It is hence suggested as major revision. Please respond the comments suggested below to revise the manuscript.

1.      It was reported that “This paper is devoted to determining the influence of the reactivity, the rotation speed, and the average diameter on the efficiency and design parameters of the carbon dioxide turbine’s flow path.” Please define the “efficiency” as a quantitative measure.

2.      Please improve the quality of Figures, except Fig. 2, as the texts in these figures are blurred.

3.      Please convert the definition of the rotating speed in tern of Hz to the unit od rad/s for the consistency with the SI units.

4.      It was reported that the losses in the blade system of stages were based on empirical dependencies for losses obtained for steam turbine profiles [18]. Please summarized these losses in a Table and comment on the limitation of the present research results owing to this assumption.

5.      With a turbine entry temperature of 1100 deg. C, the analysis of the bending and tensile stresses for the blade without internal cooling channels is not realistic. This aspect of results can be removed as such analysis seems not affecting the “efficiency” of the carbon dioxide turbine’s flow path.

6.      Typo. “Figure 5. Algorithm for designing a carbon dioxide turbomachine” Please correct Figure 5 to Figure 2.

7.      The underlined theories and their relevant equations in each milestone of Fig. 2 must be provided and explained in a systematic manner in order to clarify the analytical method adopted.

8.      Please add a nomenclature section to define all the symbols in Fig. 2 and in the manuscript with their dimensions based on SI units. An additional plot to show the blade profile with all the cited geometric factors and the flow parameters considered is required.

9.      It was reported “To maintain parameter u/cf unchanged, we need to increase the heat drop generated by each stage, ….” Please change the “heat drop generated by each stage” into “enthalpy drop across each stage”. The term “heat drop” is constantly used in the manuscript. Please correct it into enthalpy drop.

10.  Please change “Formula” into equation throughout the manuscript. Please add the reference for equation (2) or shows its derivations.

Review report of inventions-2007187

Influence of Kinematic Parameters of Carbon Dioxide Turbine on Its Efficiency and Dimensions

The present study probed into the influence of kinematic parameters of carbon dioxide turbine on its efficiency and dimensions using one-dimensional method. The major concern for ensuring the Journal quality is the description of the research method which is not sufficiently clear. The theories with their relevant equations involving in the present one-dimensional analytical method must be provided and explained in a systematic manner. It is hence suggested as major revision. Please respond the comments suggested below to revise the manuscript.

1.      It was reported that “This paper is devoted to determining the influence of the reactivity, the rotation speed, and the average diameter on the efficiency and design parameters of the carbon dioxide turbine’s flow path.” Please define the “efficiency” as a quantitative measure.

2.      Please improve the quality of Figures, except Fig. 2, as the texts in these figures are blurred.

3.      Please convert the definition of the rotating speed in tern of Hz to the unit od rad/s for the consistency with the SI units.

4.      It was reported that the losses in the blade system of stages were based on empirical dependencies for losses obtained for steam turbine profiles [18]. Please summarized these losses in a Table and comment on the limitation of the present research results owing to this assumption.

5.      With a turbine entry temperature of 1100 deg. C, the analysis of the bending and tensile stresses for the blade without internal cooling channels is not realistic. This aspect of results can be removed as such analysis seems not affecting the “efficiency” of the carbon dioxide turbine’s flow path.

6.      Typo. “Figure 5. Algorithm for designing a carbon dioxide turbomachine” Please correct Figure 5 to Figure 2.

7.      The underlined theories and their relevant equations in each milestone of Fig. 2 must be provided and explained in a systematic manner in order to clarify the analytical method adopted.

8.      Please add a nomenclature section to define all the symbols in Fig. 2 and in the manuscript with their dimensions based on SI units. An additional plot to show the blade profile with all the cited geometric factors and the flow parameters considered is required.

9.      It was reported “To maintain parameter u/cf unchanged, we need to increase the heat drop generated by each stage, ….” Please change the “heat drop generated by each stage” into “enthalpy drop across each stage”. The term “heat drop” is constantly used in the manuscript. Please correct it into enthalpy drop.

10.  Please change “Formula” into equation throughout the manuscript. Please add the reference for equation (2) or shows its derivations.

Author Response

Hello, dear reviewer! Thank you for your comments, they are really objective and will certainly enhance our manuscript. Step-by-step implementation of the comments is presented in the attachment, and in accordance with your comments, I am sending the corrected manuscript.

Reviewer 2 Report

In this study, author have presented the results of one-dimensional calculations of a flow path of the carbon dioxide turbine for the Allam cycle with a capacity of 300 MW, with an initial temperature and pressure of 1,100 °C and 30 MPa, and an outlet pressure of 3 MPa. The study was carried out by varying the rotor speed, the reactivity level and the average diameter. However, the current paper is not conditionally publishable, and should be further revised.

1. The novelty of the work must be clearly addressed and discussed, compare your research with existing research findings and highlight novelty, (compare your work with existing research findings and highlight novelty).

2. The main objective of the work must be written on the more clear and more concise way at the end of introduction section.

3. Introduction section must be written on more quality way, i.e. more up-to-date references addressed. Research gap should be delivered on more clear way with directed necessity for the conducted research work.

4. Conclusion section is missing some perspective related to the future research work, quantify main research findings.

5.The abbreviations used in this paper are very disordered. The authors should present the full name of an abbreviation when it appears the first time. Then the abbreviation can be used all through the paper.

Author Response

Hello dear reviewer! Thank you for your comments, they are really objective and will enhance our manuscript. A step-by-step implementation of the remark is presented in the attachment.

Round 2

Reviewer 1 Report

These authors have responded most of the comments satisfactorily. The symbols are self-explained in the texts. It is recommended for publication.

Reviewer 2 Report

The authors performed some revisions according to the reviewers' comments, and this paper could be considered to be published.